# Next generation insect taxonomic classification by comparing different deep learning algorithms

**Song-Quan Ong**[1]⊛*, **Suhaila Ab. Hamid**[2]⊛

**1** Institute for Tropical Biology and Conservation, Universiti Malaysia Sabah, Kota Kinabalu, Sabah, Malaysia, **2** School of Biological Sciences, Universiti Sains Malaysia, Gelugor, Pulau Pinang, Malaysia

⊛ These authors contributed equally to this work.
* songquan.ong@ums.edu.my

**Data Availability Statement:** The data underlying the results presented in the study are available from https://doi.org/10.6084/m9.figshare.19607193.v1.

## Abstract

Insect taxonomy lies at the heart of many aspects of ecology, and identification tasks are challenging due to the enormous inter- and intraspecies variation of insects. Conventional methods used to study insect taxonomy are often tedious, time-consuming, labor intensive, and expensive, and recently, computer vision with deep learning algorithms has offered an alternative way to identify and classify insect images into their taxonomic levels. We designed the classification task according to the taxonomic ranks of insects—order, family, and genus—and compared the generalization of four state-of-the-art deep convolutional neural network (DCNN) architectures. The results show that different taxonomic ranks require different deep learning (DL) algorithms to generate high-performance models, which indicates that the design of an automated systematic classification pipeline requires the integration of different algorithms. The InceptionV3 model has advantages over other models due to its high performance in distinguishing insect order and family, which is having F1-score of 0.75 and 0.79, respectively. Referring to the performance per class, Hemiptera (order), Rhiniidae (family), and *Lucilia* (genus) had the lowest performance, and we discuss the possible rationale and suggest future works to improve the generalization of a DL model for taxonomic rank classification.

## Introduction

Insects keep the planet liveable. They contribute significantly to our environment and are essential to ecological functions such as nutrient recycling, plant propagation, maintenance of the plant community, maintenance of the animal community, and food for insectivorous animals. For instance, the dipterous families Calliphoridae, Rhiniidae, and Sarcophagidae, which are ecologically important and involved intensively in nutrient recycling of organic matter [1], serve as pollinators [2] and are vectors for diseases such as cholera [3]. However, the data on changes in species diversity and abundance are insufficient. A major reason for these shortfalls

**Funding:** The author(s) received no specific funding for this work.

**Competing interests:** The authors have declared that no competing interests exist.

for insects is that available methods to study and monitor species are often tedious, time-consuming, labor-intensive, and expensive.

Deep learning (DL) algorithms with computer vison are an excellent alternative for insect taxonomists to collect insect data, especially in designing next-generation insect monitoring tools. DL algorithms consist of feature extraction and classification layers in the neural network layers [4, 5], allowing the automated system to perform end-to-end recognition tasks. DL has advantages over other machine learning algorithms, such as a support vector machine, decision tree, and logistic regression methods. For example, Motta et al. [6] leveraged the LeNet, AlexNet, and GoogLeNet convolutional neural networks in classifying six classes of field caught mosquitoes and obtained a maximum accuracy of 76.2% by GoogLeNet. Park et al. [7] utilized a variant of VGG-16, ResNet, and SqueezeNet to classify mosquito species with different postures and deformations and obtained 97% accuracy by fine-tuning the general features. Valan et al. [8] classified four datasets of insects (Diptera, Coleoptera and Plecoptera) by using VGG19 and obtained at least 90% accuracy. Ozdemir et al. [9] developed mobile apps with the deep learning algorithms VGG16 and InceptionV3 for insect order classification and achieved at least 80% average accuracy. However, most of these previous studies of DL models on insect classification were not designed to assess the capability of DL in classifying different taxonomic levels. For instance, research questions such as "What will the performance of a DL model be as the taxonomic level decreases?" and "Will a single DL architecture be sufficient to classify specimens regardless of their taxonomic levels?" remain. Since previous studies assumed that insect classification can be done according to the concept of one- size-fits-all, the most appropriate algorithm could be the solution for most classifications at the taxonomic level. We hypothesise that different algorithms for classification are needed for different taxonomic levels, because the lower the level, the closer the external morphology. For this reason, this study aims to evaluate the ability of DL models in classifying insect specimens at different taxonomic levels. We compared the performances of four DL models, InceptionV3, VGG19, MobileNetV2, and Xception, in classifying three taxonomic levels: order, family, and genus.

## Materials and methods

### Insect specimen resources and experimental design

The insect specimens were obtained from the insect collection rooms of BORNEENSIS, the Institute for Tropical Biology and Conservation (ITBC), Universiti Malaysia Sabah (UMS), and the School of Biological Sciences, Universiti Sains Malaysia (USM). Both the insect collection rooms kept a total of more than 500,000 insect specimens that were preserved and stored in a compactor at 18˚C and 40±5% relative humidity. The taxonomy of insects was identified and validated until at least the taxonomic genus rank by two taxonomists.

The experiment was designed to evaluate four state-of-the-art deep learning models in generalizing unseen and independent data of the taxonomic levels order, family, and genus. Fig 1 illustrates the overall workflow of this study. In general, the adult stage of the insect was used for image acquisition, and the annotation of the datasets was based on target output/classes of three taxonomic ranks ("*class*" in the classification task of machine learning refers to the final prediction outputs, not to be confused with the taxonomic class rank). To this end, we selected Diptera, Hemiptera and Odonata, which have distinguished morphology; we approached the families in one of the challenging orders, Diptera, which are Calliphoridae, Rhiniidae–and Sarcophagidae; for the genus, we referred to the families of Diptera as well, which are Chrysomya, Lucilia, Rhiniinae, Sarcophaga, and Stomorhina.

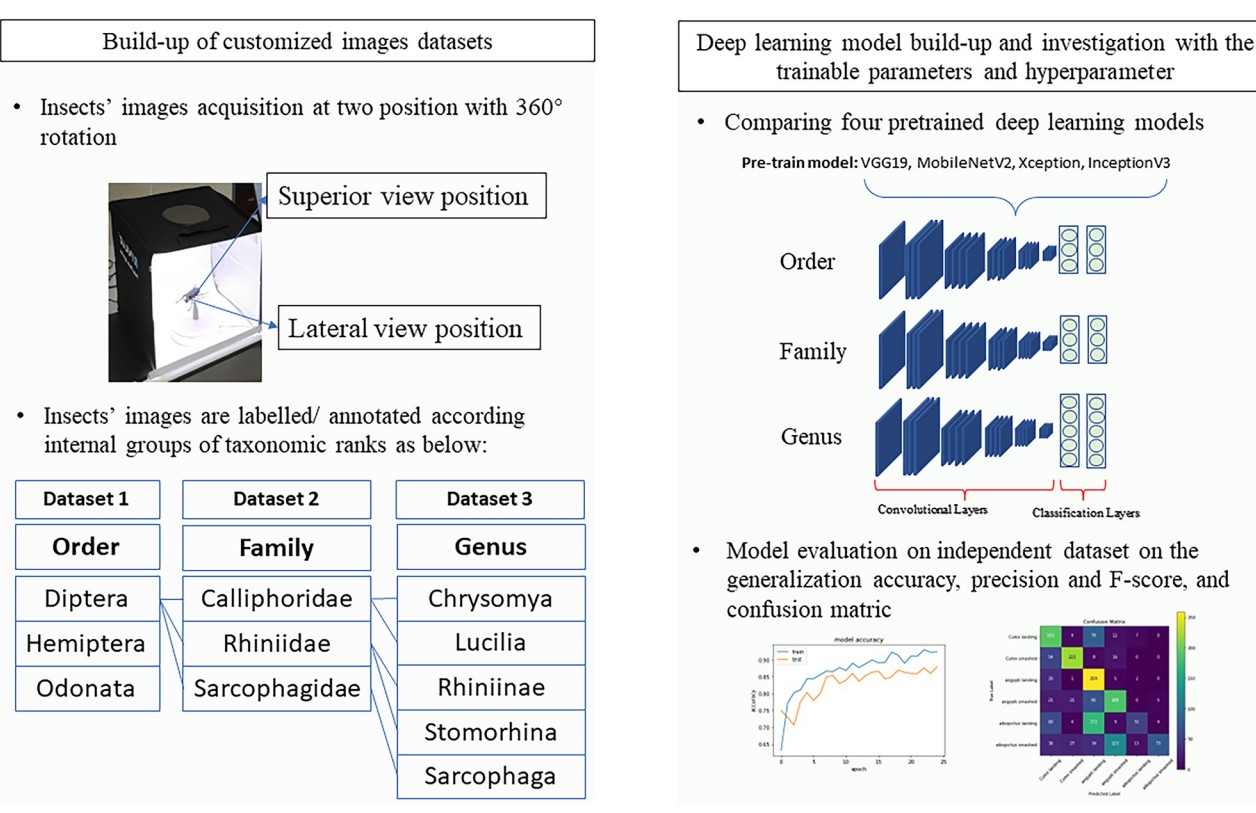

**Fig 1. Overall workflow: Stage one consists of building up three customized datasets, and stage two involves the comparison and investigation of four state-of-the-art deep learning models.**

## Data collection

The insects' images were acquired by a digital single-lens reflex (DSLR) camera (Canon EOS 50D, 15.0 MP APS-C CMOS sensor) with a Tamron 90 mm f/2.8 Di macro lens. The image acquisition process was conducted in a 30x30x30 cm photography lightbox with white light illumination. To obtain a 360° view of the specimen, the insect specimen was placed with a pin on an electronic motorized rotating plate, and the images were acquired at two levels of positions–the superior view and lateral view of the insect. For a taxonomic level, 5 to 10 specimens with different variants were used to generate the images, and at least 100 images were acquired from each specimen. As a result, at least 2000 images were acquired for one taxonomic level.

## Data preprocessing and augmentation

Most state-of-the-art deep learning architectures usually require much more training data for stable performance, with approximately 2000 images not being sufficient to train a robust model, and augmentation would be necessary [10]; therefore, we applied rotation augmentation to increase the volume of training data. We applied four-degree rotation to the images after the data were split into training, testing, and independent validation images.

The data splitting and partitioning used for training, testing, and validation of the model are described as training (70%) and testing (15%), and the prediction is carried out on an independent validation dataset (15%). The base images (0 degrees, without rotation) and all the rotated images (90, 180, and 270 degrees) used for training are not used for the testing and validation sets. For model training and evaluation, we use the Keras deep learning framework on

**Table 1. Formulas for calculating the evaluation matrics from a confusion matrix.**

| Evaluation matrics | Formulas for calculation* | Equation |
|---|---|---|
| Accuracy | $\frac{TP+TN}{TP+TN+FP+FN}$ | (1) |
| Precision | $\frac{TP}{TP+FP}$ | (2) |
| Recall | $\frac{TP}{TP+FN}$ | (3) |
| F1-score | $\frac{2\times precision\times recall}{precision+recall}$ | (4) |

*True Positive (TP); False Positive (FP); False Negative (FN); True Negative (TN)

a NVIDIA Tesla P100-PCIE GPU platform. Training is performed for 100 epochs, and the learning rates are reduced by 0.25 every 15 epochs. The standardized number of epochs for image classification was to prevent the models to overfit the training data [10].

## Deep learning model build-up

We investigated four deep learning models, MobileNetV2, InceptionV3, Xception, and VGG19, in which weights and biases were adopted for the classification of 1000 classes of the ImageNet dataset [11]. The four pretrained models were selected based on their top-5 accuracy and size (MB) of the model listed in the Keras library [12]. Xception, InceptionV3, and MobileNetV2 were selected due to their relatively smaller size and high accuracy, and VGG19 was selected as a benchmark from previous studies [7, 8]. For the architecture, the softmax layer was truncated, and the output of the model was set as the last tensor representation of the image. For the first dense layer, the input was the same as the output of the CNN, and the transformation of the data to their tensor was performed by the CNN. This study trained deep learning neural networks by using the adaptive learning rate optimization (Adam) algorithm with learning rate hyperparameters of 0.001, 0.0001, and 0.00001 to control the rate of change of the model during each step of the optimization process. The output of the optimized model was presented as the mean and standard error (SE) and used to develop 95% confidence intervals (Cis) and validate the model statistical significance by referring to the overlapping of Cis or SE bars (overlap rule for SE bars) [13]. Inference cohort classification was conducted by using new and independent datasets. The evaluation matrices used to represent the generalization of the model were accuracy, precision, recall, and F1-score. According to Zheng [14], accuracy describes the number of correct predictions over all predictions (1); precision is a measure of how many of the positive predictions made are correct true positives (2); and recall is a measure of the positive cases the classifier correctly predicted over all the positive cases in the data (3). The F1-score is a measure combining both precision and recall and is described as the harmonic mean of the two (4). Table 1 summarized the formulas used to calculate the evaluation matrix from a confusion matrix.

To prevent model overfitting, three strategies were implemented: first, we applied additional dropout regularization layers (p = 0.5) before the classification block; second, we implemented early stopping with a maximum number of iterations for which no progress was recorded; and third, we expressed multiple evaluation metrics referring to the inference of the validation dataset.

## Results

### Insect image datasets

We aim to use the deep learning (DL) models in classifying unseen data in the real world, and therefore the images that used for model training need to cover most of the angles and position

**Table 2. Number of acquired images per taxonomic rank per class.**

| | Taxonomy | Number of genera** | Number of specimens | Total images |
|---|---|---|---|---|
| Class* | Order | 3 | 7 | 6,272 |
| | Family | 5 | 25 | 6,828 |
| | Genus | 5 | 25 | 11,375 |

* The "class" in classification task of a machine learning refers to the final prediction outputs, not to confuse with the class-rank of taxonomy

** The taxonomy of insects was identified and validated until taxonomic genus rank only

views of an insect. To the best of our knowledge, a dataset that fulfills such criteria is unavailable; therefore, we created these datasets by taking insect specimen images from 5 to 6 samples using a DSLR camera with a close-up macro lens. We took approximately 60 to 100 images for each specimen on the rotating plate, which covered a 360˚ view of details of the specimen at the superior and lateral positions. Each original image has a resolution of 5184 × 3456 pixels, with 24 bits of RGB channels and 72 dpi. Through this manual image acquisition process and data augmentation, we collected the numbers of images described in Table 2. More details of creating the dataset can refer to Ong and Ahmad [15]. Fig 2 shows some of the images of the specimen, in which the camera attempts to capture most of the key morphology from different angles and positions and learn by deep learning models.

## Deep learning algorithm comparison and generalization

Our second objective of this study is to compare four deep learning (DL) models in generalizing/inferring unseen insect images according to the taxonomic levels order, family, and genus. Fig 3 shows the results of the four DL models in predicting an independent validation dataset, and Appendix I shows the confusion matrix for each of the deep learning algorithms with regard to taxonomic rank. One of the important findings of this study reveals that each taxonomic level consists of its best-performing and generalized DL model, which indicates that multiple taxonomy rank classification cannot be solved by a single DL architecture. For instance, the VGG19 model performed the best for order, InceptionV3 performed the best for


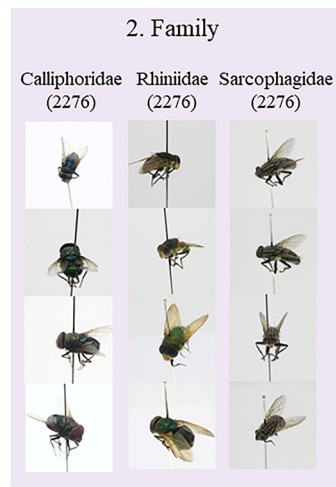
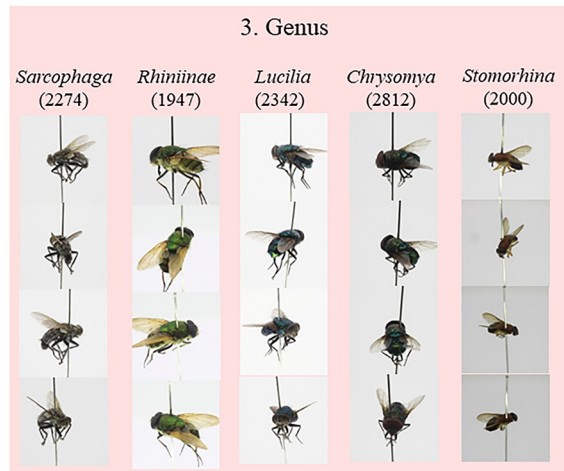

**Fig 2. Results of dataset construction:** Three datasets regarding the taxonomic levels order (three classes, Diptera, Hemiptera, Odonata), family (three classes, Calliphoridae, Rhiniidae, and Sarcophagidae), and genus (five classes, Chrysomya, Lucilia, Rhiniinae, Sarcophaga, and Stomorhina).

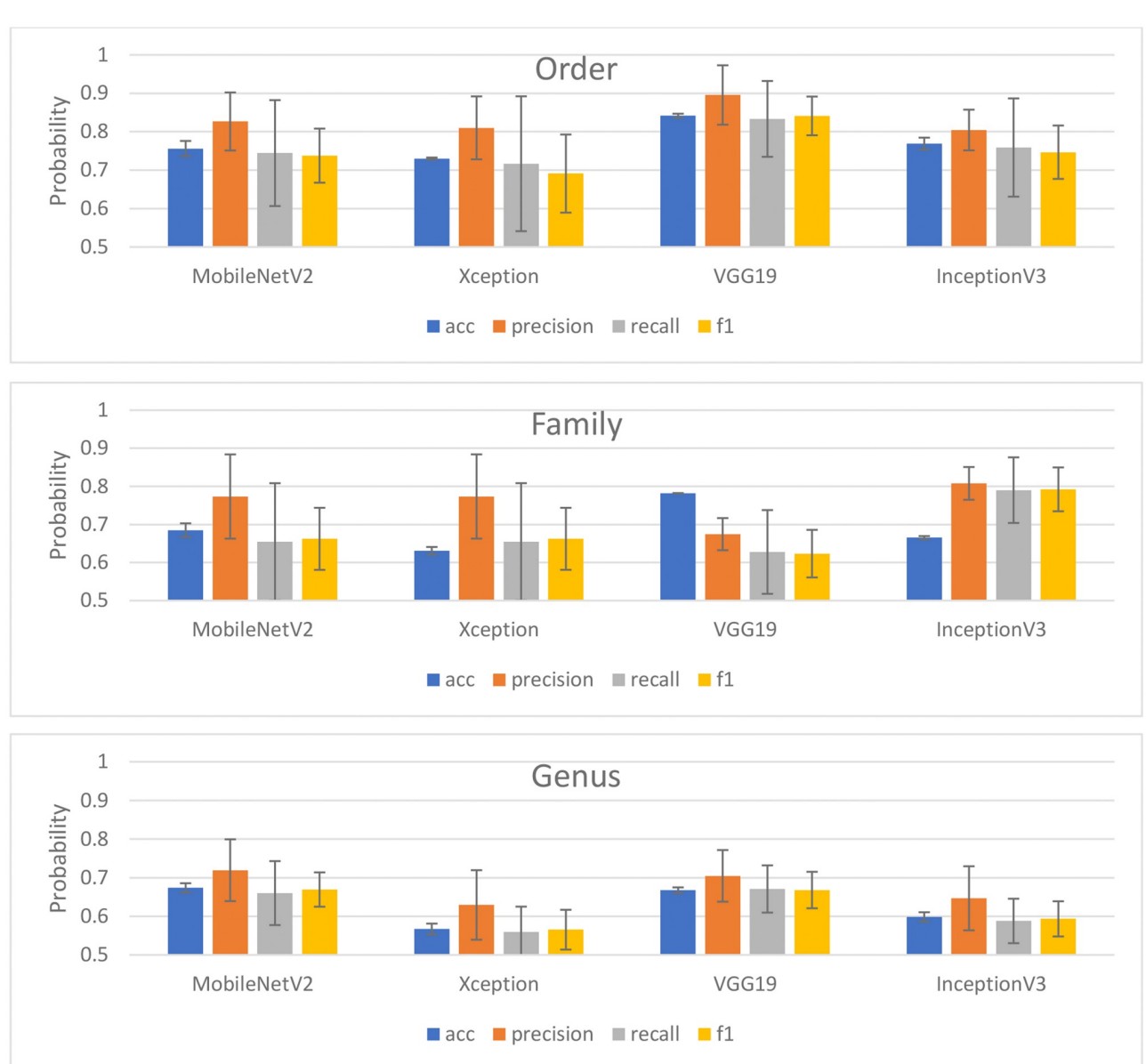

**Fig 3. Evaluation matrices of four DL models according to the respective taxonomic levels.**

family, and MobileNetV2 performed the best for genus. The inceptionV3 that having a total of 42 layers is having advantages of consistent performance from one level to another, which did not perform significantly differently when the taxonomic level was lowered from order to family, in contrast with other models that exhibited significantly lower performance when the level was lower.

We can obtain some insight from the iterative learning process of features within the layers of DL architecture, which can be observed from the learning curve and error loss of the model. Appendices II and III show the accuracy and error loss of training and the internal testing curve of the DL models based on the epochs, in which the epoch indicates the number of iterations of the entire training dataset the machine learning algorithm has completed. As seen in Appendices II and III, because early stopping was applied to prevent overfitting, the epoch

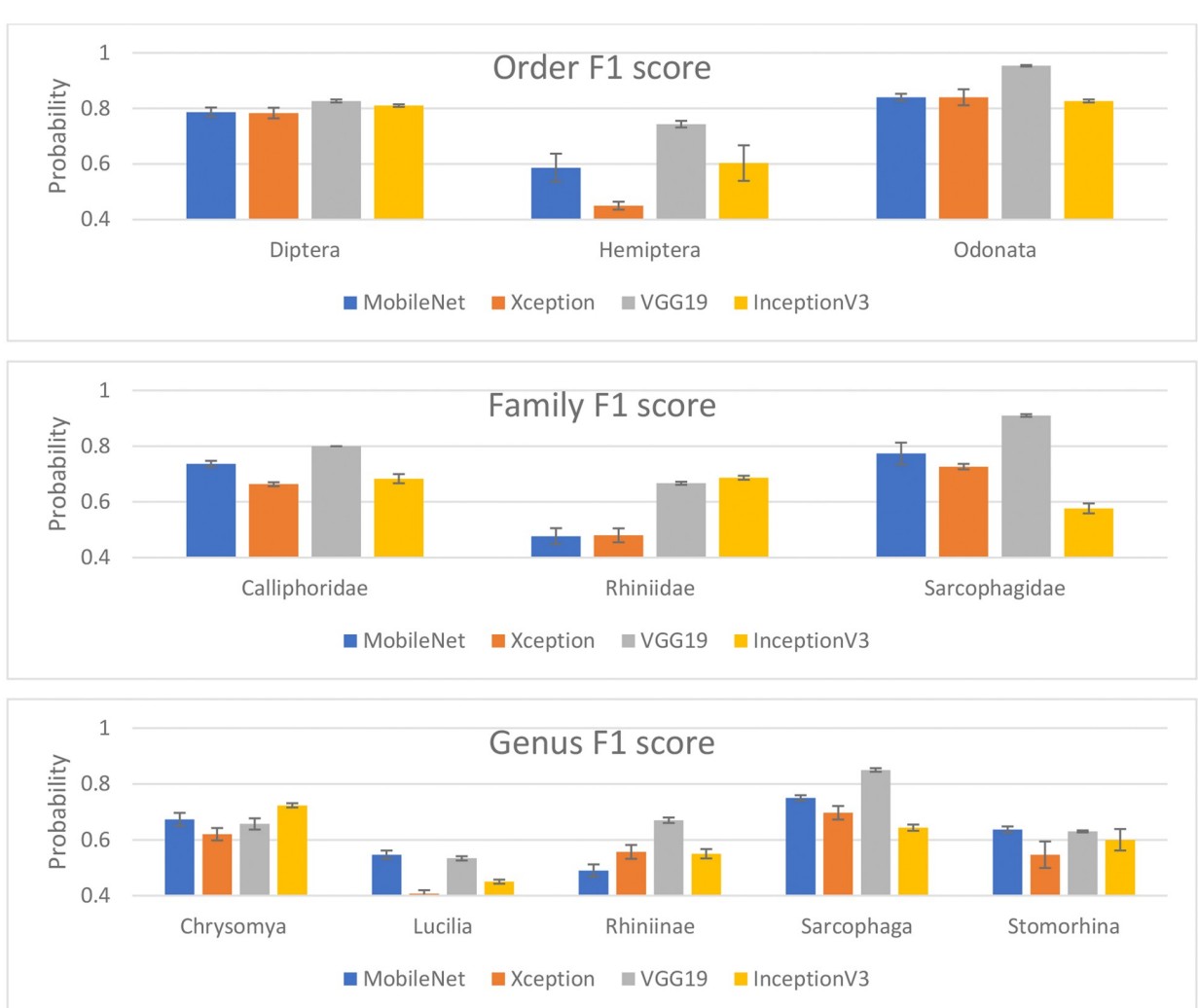

**Fig 4. Model classification based on the individual group within the level.**

could indicate the duration needed to achieve the maximum accuracy. We hypothesize that the epoch of the DL model could be longer when the taxonomic ranks are lower; however, our results revealed that the number of epochs is independent of the difficulties of taxonomic ranks. The stability of the training and testing process may affect the model performance, and our result shows that the models with a small learning rate of 0.00001 or the Xception and VGG19 models and the training and testing curves were relatively more stable.

For performance per class within the taxonomic rank, we standardized the F1 score as the assessment matrices for the comparison. Fig 4 shows the model classification based on the individual group within the level. Hemiptera had the lowest performance among the 4 studied DL models, which may be because the specimen exhibited an open wing that could be confused with Odonata. Xception and MobileNetV2 had low performance in classifying Rhiniidae, which has a metallic green body similar to Calliphoridae, and this was also observed at the genus level, where the four DL models had significantly lower performance for Lucilia and Rhiniinae.

## Discussion

Using a suitable dataset is crucial for deep learning classification tasks. Our dataset construction result is able to benchmark with previous studies, such as the study of Lytle et al. [16], who created a dataset of 9 stonefly taxa for an automated classification system called BugID; the study of Rodner et al. [17], who produced the Ecuador moth dataset; and the study of Valan et al. [8], who constructed a dataset of beetles with 3 orders. Our constructed datasets have advantages in terms of angle and position coverage–a 360° view at superior and lateral positions for image acquisition of the morphology of a specimen–and the annotation was achieved according to the taxonomic levels order, family, and genus (Fig 1). A customized dataset has also been emphasized by Goodwin et al. [18] when the recognition task was domain specific and public or when an open-source database achieved poor performance in prediction. Nevertheless, customization of the dataset always poses a challenge in terms of the cost and data size [7, 17–19] and therefore is always one of the key constraints for a DL modeling study.

When considering deep learning as the algorithm for a recognition system, we must understand the importance of the system to be used in the real world to infer unseen data. Our experimental results of the generalization of InceptionV3 and VGG19 were similar to those of the studies of Lytle et al. [16], who used a random forest algorithm with a selection operator and correctly classified 89.5% of stonefly images belonging to 9 taxa and 7 families; Valan et al. [8], who used VGG16 and obtained at least 90% internal test set accuracy on four datasets that consisted of flies, beetles and stoneflies; and Yang et al. [20], who compared InceptionV3, VGG16, and ResNet50 in classifying insect images with complicated backgrounds and concluded that InceptionV3 outperformed the other models. Nevertheless, we extended their studies by updating more comprehensive comparisons among the state-of-the-art deep learning model in the Keras library and having better performance coverage in terms of precision and F1-score. Moreover, taking note of the model characteristic such as trainable parameters versus the taxonomic level, which a decrease of parameters (VGG19 to MobileNetV2), higher the performance with lower taxonomic levels.

In addition, we determined the actual performance of the deep learning model in classifying insect external morphology according to the taxonomic rank and detailed the performance on individual classes (groups of levels). For instance, Xception and MobileNetV2 were seldom considered by previous studies in insect classification; nevertheless, MobileNetV2 has a smaller file size and is capable of classifying the insect to the species level, which was demonstrated by Ong et al. [4], who classified *Aedes aegypti* and *Aedes albopictus* mosquitoes in real time by providing key close-up morphology images as the training data. Our generalization result also supports the idea that a customized deep learning architecture is required based on taxonomic ranks.

Our results show that the model performed poorly on blow flies that have metallic bodies, such as *Lucilia* and *Rhiniinae*, but interestingly, *Chrysomya* is an exception. Therefore, to rationale that Chrysomya had significantly higher performance than Lucilia and Rhininae, we used a heatmap to visualize the region that distinguished Chrysomya and found that the identification was focused on the thorax area of the flies (Fig 5). This outcome agreed with one of the keys for the identification of *Chrysomya* and *Lucilia*, which is the relatively dark and nonmetallic thorax of *Chrysomya*, compared with *Lucilia* and *Rhininae* [21]. From the perspective of the convolutional neural network (CNN) architecture of deep learning, the feature extraction blocks in the CNN before the classification layers could consist of generic (low-level features) and specific (high-level features) features. Yosinski et al. [22] and Zeiler & Fergus [23] proposed that shallow features were generic and captured primitive patterns. Therefore, the four

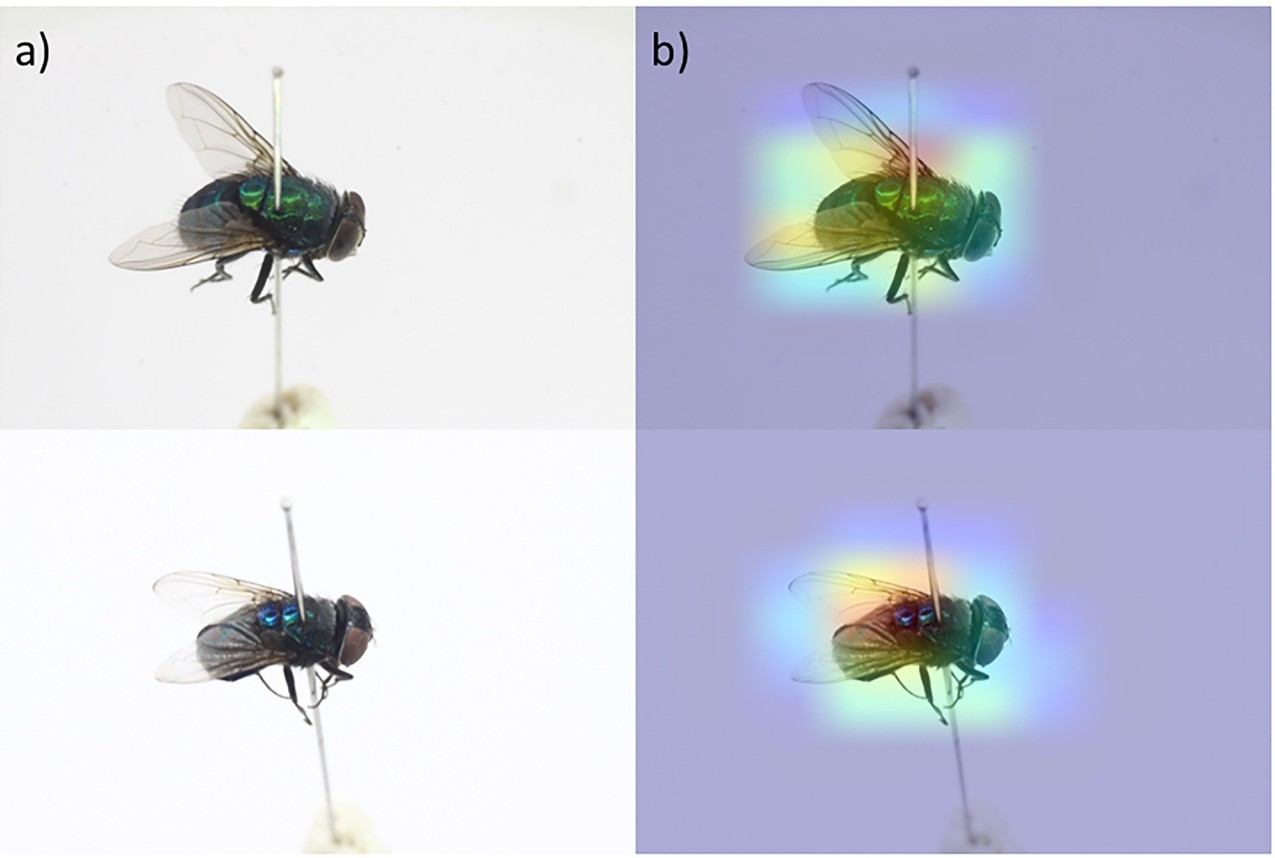

**Fig 5. Heatmap visualization of the classification region: a) original image and b) heatmap of the region used by the neural network in classification.**

models that we studied are pretrained on the images in the ImageNet dataset, which are more general and colorful than our insect images. Finally, features learned from the ImageNet dataset are generally useful for overcoming the scarcity of data, but overall fine-tuning is required by some models to capture specific features of insects and achieve better performance.

We demonstrated that a single DL architecture was not robust enough to classify different taxonomic levels of specimens. This result is crucial when future works are intended to design next-generation technologies in taxonomic classification or insect monitoring by automated recognition, and integration of different DL models may be one of the solutions. Another possible solution for automated taxonomic classification could be using other supervised machine learning models, for instance, deep recurrent neural networks (DRNN) that have the capability of fetching a previous output (result of prediction) as a new input for the current step, to self-learn the misclassify group and eventually make improvements [24]. Nevertheless, this study has some limitations in terms of image quality. First, the images used for training were museum specimens that were in good condition, and the model performance may be different when implemented/deployed on specimens caught in the field that may be damaged or contain other backgrounds or objects or a new species. Second, image data were taken in a high-resolution camera and under standardized laboratory conditions. The images were acquired by using a DSLR camera under sufficient light illumination. Therefore, images from a smartphone that has been internally processed to enhance the visualization of an image and images from the field may not be recognized by the model constructed by this dataset.

## Author Contributions

**Conceptualization:** Song-Quan Ong.

**Data curation:** Song-Quan Ong.

**Formal analysis:** Song-Quan Ong.

**Investigation:** Song-Quan Ong.

**Methodology:** Song-Quan Ong.

**Project administration:** Song-Quan Ong, Suhaila Ab. Hamid.

**Resources:** Song-Quan Ong.

**Software:** Song-Quan Ong.

**Supervision:** Suhaila Ab. Hamid.

**Validation:** Song-Quan Ong, Suhaila Ab. Hamid.

**Visualization:** Song-Quan Ong.

**Writing – original draft:** Song-Quan Ong, Suhaila Ab. Hamid.

**Writing – review & editing:** Song-Quan Ong, Suhaila Ab. Hamid.

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
