## [Decision Letter · Decision Letter 0]

3 Nov 2022

PONE-D-22-26033Next Generation Insect Taxonomic Classification by Comparing Different Deep Learning AlgorithmsPLOS ONE

Dear Dr. Ong,

Thank you for submitting your manuscript to PLOS ONE. After careful consideration, we feel that it has merit but does not fully meet PLOS ONE’s publication criteria as it currently stands. Therefore, we invite you to submit a revised version of the manuscript that addresses the points raised during the review process.

We look forward to receiving your revised manuscript.

Kind regards,

Vijayalakshmi G V Mahesh, Ph.D

Academic Editor

PLOS ONE

Journal Requirements:

- https://onlinelibrary.wiley.com/doi/10.1002/ps.7028

The text that needs to be addressed involves the Introduction and the Results sections.

In your revision ensure you cite all your sources (including your own works), and quote or rephrase any duplicated text outside the methods section. Further consideration is dependent on these concerns being addressed.

Reviewers' comments:

Reviewer's Responses to Questions

**Comments to the Author**

1. Is the manuscript technically sound, and do the data support the conclusions?

Reviewer #1: Yes

Reviewer #2: Yes

Reviewer #3: Yes

 2. Has the statistical analysis been performed appropriately and rigorously?

Reviewer #1: No

Reviewer #2: N/A

Reviewer #3: Yes

 3. Have the authors made all data underlying the findings in their manuscript fully available?

Reviewer #1: Yes

Reviewer #2: No

Reviewer #3: Yes

 4. Is the manuscript presented in an intelligible fashion and written in standard English?

Reviewer #1: Yes

Reviewer #2: Yes

Reviewer #3: Yes

 5. Review Comments to the Author

Reviewer #1: The authors have proposed a novel algorithm version of n Insect Taxonomic Classification using CNN. The following are the comments that needs to addressed in the manuscript

- Abstract and conclusion needs the accuracy/ performance evaluation results to be specified.

- The research gap and the proposed solution should be highlighted before the methodology

- The novelty of the proposed work should be highlighted.

- Is there any open source database available for for this application? is yes then the results should be obtained the the same and reported in the article.

- Discussion part needs to be elaborated and how the proposed method is efficient compared to other existing algorithms

- references should be recent (less than 5-7 years)

Reviewer #2: In this work, the authors study the classification performance of four deep CNN models (InceptionV3, VGG19, MobileNetV2 and Xception) in classifying insect images into three taxonomic levels (order, family and genus). I have only a few minor comments to improve the paper:

1. The introduction section could elaborate on the motivation for the work.

2. The authors propose that the classification pipeline must include several classification algorithms for different taxonomic ranks. It will be interesting if the authors could elaborate on the characteristic that the classification algorithm should possess to perform remarkably for each taxonomic rank.

3. How did the authors choose the optimal hyperparameters for the model?

4. Even though 2000 images may not be sufficient, it might be interesting to see how the model performs on the original dataset of ~2000 images and compare the performance with the dataset that had the rotated images as well.

5. The authors also fixed the number of training epochs at 100, which might quite low. The authors might consider increasing the number of training epochs and evaluating the performance.

Reviewer #3: The paper addresses the class classification task according to the taxonomic ranks of insects—order, family, and genus

and compared the generalization of four state-of-the-art deep convolutional neural network (DCNN) architectures. The statistical analysis for all the four Deep learning models with respect to taxonomy levels are showcased.

Model classification based on the individual group are also well depicted.

Concern: Little more detailing on preprocessing of the data and the InceptionV3 model layers could be added relevance.

 6. PLOS authors have the option to publish the peer review history of their article (what does this mean?). If published, this will include your full peer review and any attached files.

Reviewer #1: **Yes: **Rajkumar Palaniappan

Reviewer #2: No

Reviewer #3: **Yes: **ROOPA B S

---

## [Author Response · Author response to Decision Letter 0]

15 Nov 2022

Reviewer #1:

The authors have proposed a novel algorithm version of Insect Taxonomic Classification using CNN. The following are the comments that needs to addressed in the manuscript

- Abstract and conclusion needs the accuracy/ performance evaluation results to be specified.

Response: Thank you for your comment. The performance evaluation results of F1-score for InceptionV3 has been added in the abstract as the sentences of "The InceptionV3 model has advantages over other models due to its high performance in distinguishing insect order and family, which is having F1-score of 0.75 and 0.79, respectively"

- The research gap and the proposed solution should be highlighted before the methodology

Response: Thank you for your comment. Research gap and hypothesis were added in the end of introduction before the methodology in line 82 - 95.

"However, most of these previous studies of DL models on insect classification were not designed to assess the capability of DL in classifying different taxonomic levels. For instance, research questions such as “What will the performance of a DL model be as the taxonomic level decreases?” and “Will a single DL architecture be sufficient to classify specimens regardless of their taxonomic levels?” remain. Since previous studies assumed that insect classification can be done according to the concept of one- size-fits-all, the most appropriate algorithm could be the solution for most classifications at the taxonomic level. We hypothesise that different algorithms for classification are needed for different taxonomic levels, because the lower the level, the closer the external morphology. For this reason, this study aims to evaluate the ability of DL models in classifying insect specimens at different taxonomic levels. We compared the performances of four DL models, InceptionV3, VGG19, MobileNetV2, and Xception, in classifying three taxonomic levels: order, family, and genus."

- The novelty of the proposed work should be highlighted.

Response: We have restructured the sentences and emphasis of the novelty of the study, which are 

1. Customised datasets (line 191 to 193)

2. No one-size-fits-all model, and each taxa levels is having their own best performed algorithm (line 221)

- Is there any open-source database available for for this application? is yes then the results should be obtained the the same and reported in the article.

Response: Yes, there is a open source of dataset available in [15]. We have mentioned the dataset in line 197 and data availability.

- Discussion part needs to be elaborated and how the proposed method is efficient compared to other existing algorithms

Response: Thank you for your comment. We elaborated how our result is more effective compared to other studies in line 281-283, where describing our result is more comprehensive and having better performance coverage including the F1-score and precision.

- references should be recent (less than 5-7 years)

Response: Thank you for your comment. We updated the reference [12] (the one reference with older than 7 years) into: 

Tang L, Zhang H, Zhang B. A note on error bars as a graphical representation of the variability of data in biomedical research: choosing between standard deviation and standard error of the mean. Journal of Pancreatology. 2019 Sep 1;2(03):69-71. 

Which published in 2019 and having more compherasive discussion on the error bar that we used as the stat tool in this study.

Reviewer #2:

In this work, the authors study the classification performance of four deep CNN models (InceptionV3, VGG19, MobileNetV2 and Xception) in classifying insect images into three taxonomic levels (order, family and genus). I have only a few minor comments to improve the paper:

1. The introduction section could elaborate on the motivation for the work.

Response: Thank you for your comment. Motivation, research gap and hypothesis were added in the end of introduction in line 82 - 95.

"However, most of these previous studies of DL models on insect classification were not designed to assess the capability of DL in classifying different taxonomic levels. For instance, research questions such as “What will the performance of a DL model be as the taxonomic level decreases?” and “Will a single DL architecture be sufficient to classify specimens regardless of their taxonomic levels?” remain. Since previous studies assumed that insect classification can be done according to the concept of one- size-fits-all, the most appropriate algorithm could be the solution for most classifications at the taxonomic level. We hypothesise that different algorithms for classification are needed for different taxonomic levels, because the lower the level, the closer the external morphology. For this reason, this study aims to evaluate the ability of DL models in classifying insect specimens at different taxonomic levels. We compared the performances of four DL models, InceptionV3, VGG19, MobileNetV2, and Xception, in classifying three taxonomic levels: order, family, and genus."

2. The authors propose that the classification pipeline must include several classification algorithms for different taxonomic ranks. It will be interesting if the authors could elaborate on the characteristic that the classification algorithm should possess to perform remarkably for each taxonomic rank.

Response: Thank you for your comment. We elaborate more on the algorithm characteristic in the section of discussion, where taking note of the model characteristic such as trainable parameters versus the taxonomic level, which a decrease of parameters (VGG19 to MobileNetV2), higher the performance with lower taxonomic levels.

3. How did the authors choose the optimal hyperparameters for the model?

Response: The optimal hyperparameters were chosen manually by comparing different learning rate and two of standard optimisers. We have described the process of studying the optimization of model in line 156-159 "This study trained deep learning neural networks by using the adaptive learning rate optimization (Adam) algorithm with learning rate hyperparameters of 0.001, 0.0001, and 0.00001 to control the rate of change of the model during each step of the optimization process.

4. Even though 2000 images may not be sufficient, it might be interesting to see how the model performs on the original dataset of ~2000 images and compare the performance with the dataset that had the rotated images as well.

Response: Thank you for your comment. Comparison of model performance by using original image number and data augmented number is not the objective of this study, therefore we added one reference [10] to justify the needs of augmenting the data before the deep model development.

Shorten C, Khoshgoftaar TM. A survey on image data augmentation for deep learning. Journal of big data. 2019 Dec;6(1):1-48.

5. The authors also fixed the number of training epochs at 100, which might quite low. The authors might consider increasing the number of training epochs and evaluating the performance.

Response: We applied early stop mechanism (Appendices II and III) to prevent the overfitting for the image classification. In other words, higher epochs could lead to the issue of overfitting, we further justified the epochs number with an additional reference - A survey on Image Data Augmentation for Deep Learning (especially Fig 1) 

Shorten C, Khoshgoftaar TM. A survey on image data augmentation for deep learning. Journal of big data. 2019 Dec;6(1):1-48

Reviewer #3:

The paper addresses the class classification task according to the taxonomic ranks of insects—order, family, and genus and compared the generalization of four state-of-the-art deep convolutional neural network (DCNN) architectures. The statistical analysis for all the four Deep learning models with respect to taxonomy levels are showcased. Model classification based on the individual group are also well depicted. Concern: Little more detailing on preprocessing of the data and the InceptionV3 model layers could be added relevance

Response: Thank you for your comment. We added more details on the preprocessing of data in line 140-141 "The base images (0 degrees, without rotation) and all the rotated images (90, 180, and 270 degrees) used for training are not used for the testing and validation sets.", and InceptionV3 model layers in line 226 - "For instance, the VGG19 model performed the best for order, InceptionV3 performed the best for family, and MobileNetV2 performed the best for genus. The inceptionV3 that having a total of 42 layers is having advantages of consistent performance from one level to another, which did not perform significantly differently when the taxonomic level was lowered from order to family, in contrast with other models that exhibited significantly lower performance when the level was lower.

Thank you very much for the valuable feedback and comment.

Best regards,

Song-Quan Ong

---

## [Decision Letter · Decision Letter 1]

1 Dec 2022

Next Generation Insect Taxonomic Classification by Comparing Different Deep Learning Algorithms

PONE-D-22-26033R1

Dear Dr. Ong,

We’re pleased to inform you that your manuscript has been judged scientifically suitable for publication and will be formally accepted for publication once it meets all outstanding technical requirements.

Kind regards,

Vijayalakshmi G V Mahesh, Ph.D

Academic Editor

PLOS ONE

Additional Editor Comments (optional):

Reviewers' comments:

Reviewer's Responses to Questions

**Comments to the Author**

1. If the authors have adequately addressed your comments raised in a previous round of review and you feel that this manuscript is now acceptable for publication, you may indicate that here to bypass the “Comments to the Author” section, enter your conflict of interest statement in the “Confidential to Editor” section, and submit your "Accept" recommendation.

Reviewer #1: All comments have been addressed

Reviewer #3: All comments have been addressed

2. Is the manuscript technically sound, and do the data support the conclusions?

Reviewer #1: Yes

Reviewer #3: Yes

3. Has the statistical analysis been performed appropriately and rigorously? 

Reviewer #1: Yes

Reviewer #3: Yes

4. Have the authors made all data underlying the findings in their manuscript fully available?

Reviewer #1: Yes

Reviewer #3: Yes

5. Is the manuscript presented in an intelligible fashion and written in standard English?

Reviewer #1: Yes

Reviewer #3: Yes

6. Review Comments to the Author

Reviewer #1: Title:Next Generation Insect Taxonomic Classification by Comparing Different Deep Learning Algorithms

The author's have addressed all the comments raised and the proposed method is novel .

Reviewer #3: All the comments are addressed.

VGG19 used is an advanced CNN model capable of complex classification tasks. This deep model is showcased for the taxonomy rank classification with appropriate classification scores and statistical analysis.

7. PLOS authors have the option to publish the peer review history of their article (what does this mean?). If published, this will include your full peer review and any attached files.

Reviewer #1: **Yes: **RAJKUMAR PALANIAPPAN

Reviewer #3: **Yes: **ROOPA B S

---

## [Editor Report · Acceptance letter]

5 Dec 2022

PONE-D-22-26033R1 

Next Generation Insect Taxonomic Classification by Comparing Different Deep Learning Algorithms 

Dear Dr. Ong:

I'm pleased to inform you that your manuscript has been deemed suitable for publication in PLOS ONE. Congratulations! Your manuscript is now with our production department. 

Kind regards, 

on behalf of

Dr. Vijayalakshmi G V Mahesh 

Academic Editor

PLOS ONE